# Genetic analysis of body weight in wild populations of medaka fish from different latitudes

**Tamiris I. Yassumoto[1], Mana Nakatsukasa[1,2], Atsushi J. Nagano[3], Masaki Yasugi[4¤a], Takashi Yoshimura[1,2,5]\*, Ai Shinomiya[2¤b]\***

**1** Laboratory of Animal Integrative Physiology, Graduate School of Bioagricultural Sciences, Nagoya University, Nagoya, Japan, **2** Division of Seasonal Biology, National Institute for Basic Biology, National Institutes of Natural Sciences, Okazaki, Aichi, Japan, **3** Faculty of Agriculture, Ryukoku University, Otsu, Shiga, Japan, **4** Laboratory of Neurophysiology, National Institute for Basic Biology, National Institutes of Natural Sciences, Okazaki, Aichi, Japan, **5** Institute of Transformative Bio-Molecules (WPI-ITbM), Nagoya University, Nagoya, Japan

¤a Current address: Faculty of Engineering, Utsunomiya University, Utsunomiya, Tochigi, Japan
¤b Current address: Exploratory Research Center on Life and Living Systems, National Institutes of Natural Sciences, Okazaki, Aichi, Japan
\* takashiy@agr.nagoya-u.ac.jp (TY); aishinom@nibb.ac.jp (AS)

**Data Availability Statement:** Raw sequence data were deposited in the DDBJ Sequence Read Archive (DRA) (https://www.ddbj.nig.ac.jp/dra/

## Abstract

The genetic bases of growth and body weight are of economic and scientific interest, and teleost fish models have proven useful in such investigations. The *Oryzias latipes* species complex (medaka) is an abundant freshwater fish in Japan and suitable for genetic studies. We compared two wild medaka stocks originating from different latitudes. The Maizuru population from higher latitudes weighed more than the Ginoza population. We investigated the genetic basis of body weight, using quantitative trait locus (QTL) analysis of the $F_2$ offspring of these populations. We detected one statistically significant QTL for body weight on medaka chromosome 4 and identified 12 candidate genes that might be associated with body weight or growth. Nine of these 12 genes had at least one single nucleotide polymorphism that caused amino acid substitutions in protein-coding regions, and we estimated the effects of these substitutions. The present findings might contribute to the marker-assisted selection of economically important aquaculture species.

## Introduction

Growth and body weight are economically important traits in the livestock industry and in aquaculture. Such traits involve complex physiological processes that are controlled by various environmental and genetic factors. Quantitative trait locus (QTL) mapping and marker-assisted selection for economic traits, including growth and body weight in aquaculture, have recently been conducted in several studies using molecular markers such as microsatellites and single nucleotide polymorphisms (SNP) [1–7].

Body weight depends not only on growth traits but also on body composition and metabolism. Genome-wide association studies (GWAS) of body mass index (BMI) over the past decade have associated several hundred SNPs with body weight and obesity [8,9].

index.html) under the accession numbers DRR226810-DRR226964.

**Funding:** This work was supported by a JSPS KAKENHI Grant-in-Aid for Scientific Research (C) (15K07163, 19K06785) to AS, Grants-in-Aid for Specially Promoted Research (26000013), and for Scientific Research (S) (19H05643) to TY, and the Human Frontier Science Program (RGP0030/2015) to TY. There was no additional external funding received for this study. The funders had no role in the study design, data collection and analysis, decision to publish, or preparation of the manuscript.

**Competing interests:** No authors have competing interests.

Animal models play essential roles in most aspects of medicine. Diet-induced obesity in zebrafish and mammalian obesity are pathophysiologically similar; thus, the roles of genes associated with visceral adiposity have been examined in zebrafish models of human obesity [10,11]. Understanding the genetic basis of body weight in teleost fish models could help deepen the medical understanding of obesity.

Ecological profiles of body size are distributed within species according to the Bergmann's rule, which states that animals living at high latitudes are generally larger than those living in low latitudes [12]. Some exceptions exist, but the findings of several studies are in line with the Bergmann's rule and its applicability in many types of mammals [13], birds [14,15], and ectothermic vertebrate and invertebrate taxa [16]. However, the underlying genetic mechanisms that result in a body size cline remain obscure.

Medaka are small freshwater fish that are native to Japan, Korea, and China. Japanese wild populations of the *Oryzias latipes* species complex are widely distributed from high to low latitudes throughout the Japanese archipelago. Previous phylogeographic studies using allozymes, mitochondrial DNA (mtDNA) sequences, and genome-wide SNP analysis revealed that Japanese wild medaka comprise Northern and Southern Japanese populations [17–20]. The average rate of SNPs between two inbred strains derived from the two populations is 3.4% [21]. Inbred strains and wild stocks of medaka originating from Japanese wild populations are currently available through the National BioResource Project (NBRP) (https://shigen.nig.ac.jp/medaka/top/top.jsp). Phenotypic variations between these two populations have been described for several traits, including brain [22] and craniofacial morphology [23], body color and sexual dimorphism [24], vertebral regionalization and number [25], aggressiveness [26], startle behavior [27], and male-specific ossified processes and sex steroid levels [28,29]. Several QTL have been detected by focusing on these phenotypic variations using genetic analyses based on draft [21] and updated medaka genome sequences (http://utgenome.org/medaka_v2/#!Top.md). We investigated two medaka wild stocks originating from different latitudes. Body weight was higher in the Maizuru population than in the Ginoza population from high and low latitudes, respectively. We investigated the genetic basis of body weight via QTL analysis of the $F_2$ offspring of these medaka populations. We detected one statistically significant QTL for body weight on chromosome 4 and assessed candidate genes located within that QTL region.

## Materials and methods

### Ethics statement

The Animal Experiment Committee at the National Institutes of Natural Sciences, Japan approved the study protocol (14A108, 15A047). The medaka used in these experiments were treated according to the animal experiment guidelines of the National Institutes of Natural Sciences, Japan.

### Animals

The NBRP Medaka (https://shigen.nig.ac.jp/medaka/) supplied adult medaka ($G_0$ generations) from stocks originating from wild North and South Japanese populations at Maizuru City (Maizuru stock; strain ID, WS215) located at 35° 28′ N 135° 23′ E, Kyoto Prefecture) and Ginoza Village (Ginoza stock; strain ID, WS255) located at 26° 28′ N 127° 58′ E, Kunigamigun, Okinawa Prefecture), respectively. We then raised several generations of these fish in the laboratory. The Maizuru and Ginoza stocks were crossed to collect the $G_1$ generations, respectively. Then, Maizuru females were mass-mated with Ginoza males and vice versa to obtain the $F_1$ offspring, which was used to analyze body weight. Five Maizuru × Ginoza pairs were mated to

generate the $F_2$ offspring for QTL mapping. Three of these pairs comprised Ginoza females and Maizuru males and two comprised Maizuru females and Ginoza males.

Two to four weeks after hatching, all the generations were transferred outdoors between June and July, 2014, and maintained for 1 year under natural temperature and photoperiod at the outdoor experimental field of the National Institute of Basic Biology (34˚ 57′ N 137˚ 9′ E) in Okazaki, Aichi, Japan. Between June and July 2015, the fish were transferred to experimental aquariums and maintained in water circulation systems at 26˚C ± 1˚C for 1 month. The fish were fed with artificial dry food twice daily. Body weight was analyzed in 10 Ginoza $G_1$, 26 Maizuru $G_1$, 15 $F_1$ hybrid, and 126 $F_2$ fish euthanized with 0.05% 3-aminobenzoic acid ethyl ester methanesulfonate salt (MS222). Water was removed from the fish by blotting them with paper towels; then, each fish was weighed. Thereafter, the fish were frozen in liquid nitrogen and stored in a deep freezer (-80˚C).

## Restriction site-associated DNA sequencing (RAD-Seq) and SNP markers

Genomic DNA was extracted from muscle tissue using DNeasy Blood & Tissue Kits (Qiagen GmbH, Hilden, Germany) according to the manufacturer's instructions. The concentration of DNA was determined using a Qubit 3.0 fluorometer (Thermo Fisher Scientific Waltham, MA, USA). Genomic DNA (40 ng) from each sample was digested using two restriction enzymes, BglII and EcoRI, ligated with a Y-shaped adaptor, and amplified by polymerase chain reaction (PCR) using KAPA HiFi HS ReadyMix (Kapa Biosystems Inc., Wilmington, MA, USA). Fragments (~300–360 bp) were selected using E-Gel Size Select (Life Technologies, Carlsbad, CA, USA). Details of the library preparation method are described elsewhere [30]. The fragments were sequenced on a HiSeq 2500 platform (Illumina Inc., San Diego, CA, USA) in 50-bp single-end mode. We conducted RAD-Seq in 10 parent fish from the 5 Maizuru × Ginoza pairs, 3–4 from the $F_1$ generation of each Maizuru × Ginoza pair, and 126 of the $F_2$ generation. The reads were quality filtered using Trimmomatic [31] under the following parameters: trimmomatic-0.32.jar SE -threads 8 -phred33 ILLUMINACLIP TruSeq3-PE-2.fa:2:30:10 LEADING:19 TRAILING:19 SLIDINGWINDOW:30:20 AVGQUAL:20 MINLEN:51. The trimmed reads were mapped to the draft genome of the Hd-rR inbred medaka strain (v. 2.2.4, http://utgenome.org/medaka_v2/#!Assembly.md), and SNPs were called using the Stacks pipeline [32]. We identified RAD tags with a homozygous genotype in all Maizuru and all Ginoza and those that had different alleles between all Maizuru and all Ginoza parents for SNP marker selection. Among these markers, we selected those that were heterozygous in all the $F_1$ individuals and genotyped in > 80% of the 126 $F_2$ fish. Finally, we selected 371 RAD markers for QTL analysis (S1 Table). Genetic distances (cM) involving each chromosome were calculated using the Kosambi map function [33].

Raw sequence data were deposited in the DDBJ Sequence Read Archive (DRA) (https://www.ddbj.nig.ac.jp/dra/index.html) under the accession numbers DRR226810-DRR226964.

## Analysis of QTL

Quantitative trait loci associated with body weight were mapped in the 126 $F_2$ fish, and simple interval mapping [34] proceeded using R/qtl software [35,36]. Genome-wide significant (5%) and suggestive (10%) thresholds of a single QTL were determined by 1000 permutation tests. Bayesian credible intervals (95% CI) were computed using the R/qtl function. Physical lengths of credible intervals (Mb) were predicted by extending the physical position of the nearest flanking markers.

## Analyses of SNPs in the candidate genes

We selected genes within the 95% CI that were positioned according to the NCBI ASM223467v1 (GCF_002234675.1) assembly and associated with growth, body weight, and obesity on the basis of a literature search. We sequenced the selected candidate genes for the Maizuru and Ginoza fish and cataloged the SNPs located on their coding regions (S2 Table). We then analyzed these polymorphisms using GENETYX software (version 13, GENETICS Inc., Tokyo, Japan) to detect variants that could cause amino acid substitutions. We only considered polymorphisms in which all eight sequenced Ginoza and Maizuru individuals (n = 4 each) were homozygous for the allele. We then analyzed the amino acid substitutions using Protein Variation Effect Analyzer (PROVEAN) v1.1 to estimate their functional effects on the encoded protein [37]. A non-synonymous amino acid substitution with a potential functional effect was found on one of the genes (*sned1*), and its protein sequence (accession number XP_011471983.1) was compared with those of two teleost fishes, *Gasterosteus aculeatus* (stickleback; ENSGACG00000003698) and *Danio rerio* (zebrafish; XP_017212114.1), *Mus musculus* (mice; XP_006529380.1), and *Homo sapiens* (humans; XP_011509233) using the sequence alignment tool, ClustalW (version 2.1, DNA Data Bank of Japan).

## Results

### Body weight variation

Fig 1 shows that body weight was significantly higher in the Maizuru than in the Ginoza ($p < 0.01$, Student $t$-test), and Fig 2 shows a broader range of body weight distribution in the $F_2$ generation than in the parental populations.

### Identification of body weight QTL

A genetic map was constructed using 371 SNP markers obtained by RAD-seq (Fig 3). The total length of the genetic map of the 24 chromosomes was 1496.32 cM, and the average calculated interval between each marker was 4.51 cM (Table 1). Physical lengths were determined based on reference medaka genome data. The total physical length was 676.74 Mb, and the average interval between each marker was 1.75 Mb (Table 1).

A QTL analysis of the 126 $F_2$ fish identified a statistically significant QTL region on the distal arm of chromosome 4, with a maximum LOD of 4.14 (Figs 3 and 4). The closest SNP marker was 56900. We calculated the mean body weight of the $F_2$ individuals that were homozygous for the Maizuru and Ginoza alleles of the closest marker and the heterozygous fish. The mean body weight was higher among individuals that carried a homozygous or heterozygous Maizuru allele than among those that carried a homozygous Ginoza allele (Fig 4C). We confirmed a linkage between an amplified fragment length polymorphism marker located on the distal position of chromosome 4 (0.68 Mb, S3 Table) and the SNP marker 56900 (4.7 cM) by genotyping using PCR.

### Differences in the amino acid sequences of the candidate genes between Maizuru and Ginoza

The 95% Bayesian CI of the QTL was 0–7 cM (Fig 4B), and the physical location of the CI estimated from the genetic and physical positions of the markers 56900 (0 cM, 2.38 Mb) and 55681 (17.7 cM, 11.01 Mb) that were the closest to the QTL (Fig 3B) was 0–5.74 Mb. Among the 141 genes encoding proteins within the CI, we identified 12 that are reportedly associated with body weight or growth. Nine of these genes had at least one SNP that caused substitutions of amino acids in the coding regions (Table 2). We estimated, using PROVEAN, that most of

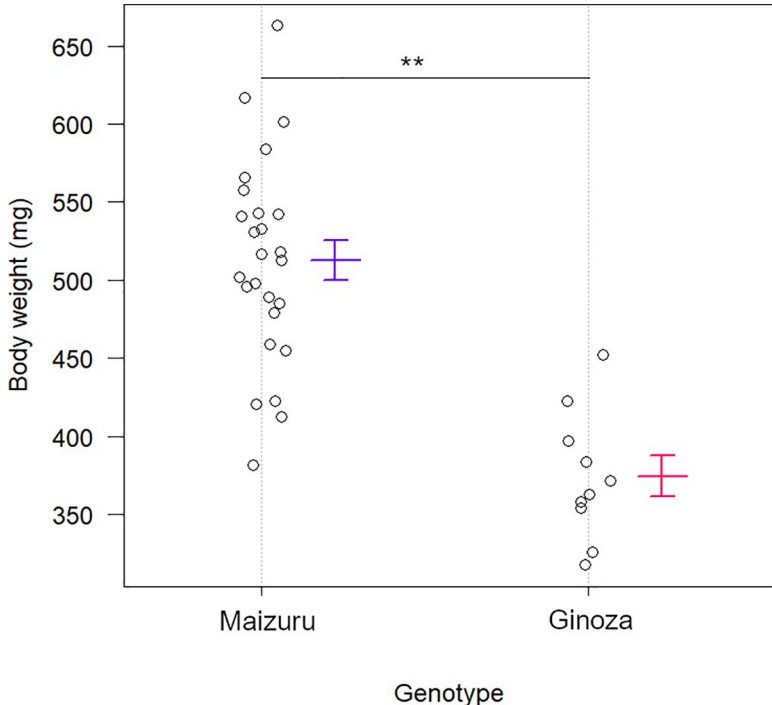

**Fig 1. Mean body weight of Maizuru (n = 26) and Ginoza (n = 10).** Data are shown as the means and standard errors (SE). Body weight was significantly higher in Maizuru than in Ginoza stock (512.6 vs. 374.7 mg; $^{**}p < 0.01$; Student $t$-tests).

these amino acid substitutions exerted neutral protein functional effects. However, one substitution in the protein SNED1 encoded by the gene *sned1* at G1013S appeared to have a deleterious (significantly different function) effect with a PROVEAN score of -2.648 (cutoff, -2.5). The amino acid glycine at 1013 in Ginoza SNED1 is conserved across other vertebrates, such as teleost, stickleback and zebrafish, as well mice and humans (Fig 5). In contrast, serine, which substituted for glycine in Maizuru SNED1, was not found in any other analyzed species.

## Discussion

Medaka fish found at various latitudes provide an excellent model for investigating the genetic basis of body weight. Therefore, we compared two wild medaka stocks from Maizuru and Ginoza at different latitudes. Fish from the parental populations and the $F_1$ and $F_2$ generations were reared under the same environmental conditions to avoid the effects of plastic responses to temperature and other variables such as food availability during growth. Mean body weight differed significantly between the Maizuru and Ginoza individuals (Fig 1), reflecting the involvement of genetic components in the determination of body size. The Bergmann's rule is currently defined as a within-species tendency for body size to increase as latitude increases [12]. Our results conformed to Bergmann's rule, as body weight was greater among the Northern Maizuru population than the Southern Ginoza population.

The identification of genes that regulate complex multigenic traits such as growth and body weight has proven challenging. Over 6000 genes are considered to influence body weight in mice [38]. Multiple QTL regions are associated with body weight in Atlantic salmon (*Salmo salar*) [2]. Our QTL analysis identified a significant QTL on chromosome 4 (peak LOD, 4.14) (Fig 3), and the effects of the allele of the marker with the highest score supported the

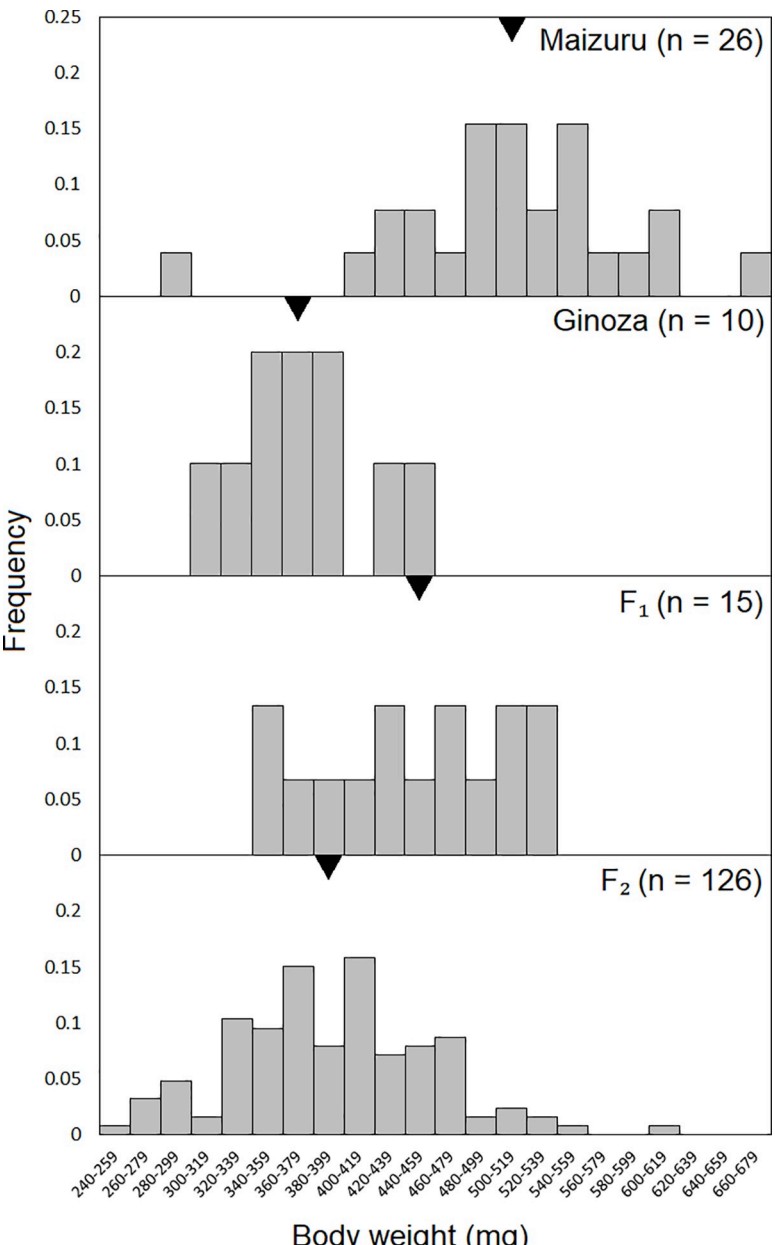

**Fig 2. Distribution of body weight of the Ginoza, Maizuru, F₁, and F₂ fish.** The Y axis indicates the frequency; the X axis indicates the body weight categories at 20-mg intervals. Black triangles indicate the means.

phenotype found in the parental strains (the body weight was higher for the Maizuru allele than the Ginoza allele in the $F_2$ medaka). This explains 14% of the variance.

We identified 12 candidate genes involved in body weight and growth regulation within the significant QTL region. The genes *cilp2*, expressing cartilage intermediate layer protein 2, and *sned1*, expressing sushi, nidogen and EGF-like domain-containing protein 1, are associated with BMI in humans [39]; *mef2b* (myocyte-specific enhancer factor 2B), *rfxank* (regulatory factor X-associated ankyrin-containing protein), and *rab6b* (Ras-related protein Rab-6B) are associated with body weight and growth traits in sheep [40–42]; *sgcb* (sarcoglycan beta) is

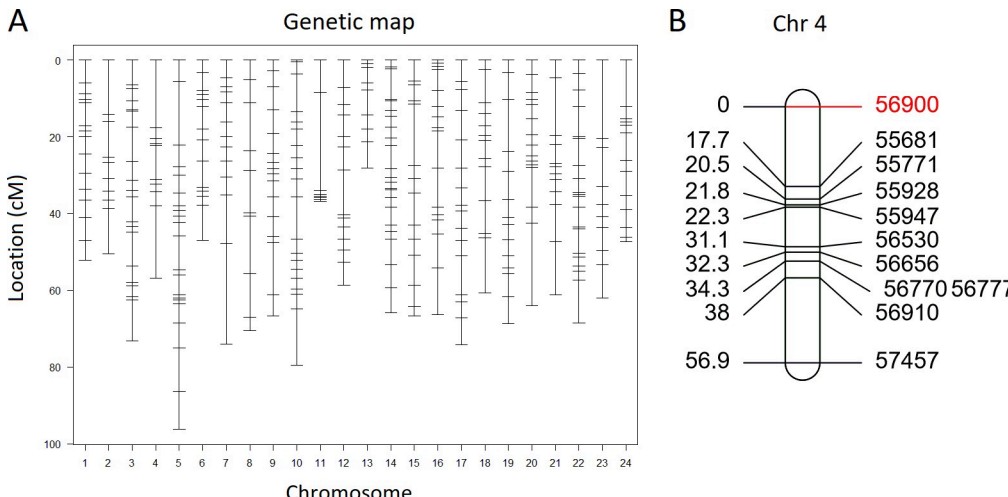

**Fig 3. Genetic map of the single nucleotide polymorphism (SNP) markers.** (A) Marker distribution across the 24 medaka chromosomes. (B) Genetic map of chromosome 4 show the marker names and locations (cM). Markers with the highest logarithm of odds (LOD) scores in the QTL analysis are shown in red.

**Table 1. Summary of markers used in the present study.**

| Chromosome no. | No. of markers | Genetic length (cM) | Average marker interval (cM) | Physical length (Mb) | Average marker interval (Mb) |
|---|---|---|---|---|---|
| 1 | 16 | 52.23 | 3.48 | 34.38 | 2.05 |
| 2 | 10 | 50.5 | 5.61 | 22.52 | 2.1 |
| 3 | 22 | 73.18 | 3.48 | 37.51 | 1.68 |
| 4 | 11 | 56.87 | 5.69 | 29.86 | 2.75 |
| 5 | 21 | 96.32 | 4.82 | 32.35 | 1.61 |
| 6 | 14 | 47.04 | 3.62 | 28.5 | 1.32 |
| 7 | 15 | 74.12 | 5.29 | 33.17 | 1.92 |
| 8 | 10 | 70.61 | 7.84 | 26.12 | 2.79 |
| 9 | 16 | 66.73 | 4.45 | 33.05 | 1.72 |
| 10 | 21 | 79.51 | 3.98 | 31.09 | 1.52 |
| 11 | 15 | 36.82 | 2.63 | 23.49 | 1.65 |
| 12 | 16 | 58.75 | 3.92 | 24.39 | 1.25 |
| 13 | 10 | 28.19 | 3.13 | 26.89 | 1.46 |
| 14 | 24 | 65.83 | 2.86 | 28.27 | 1.01 |
| 15 | 14 | 66.65 | 5.13 | 29.59 | 1.9 |
| 16 | 19 | 66.33 | 3.68 | 29.84 | 1.25 |
| 17 | 16 | 74.15 | 4.94 | 31.74 | 1.92 |
| 18 | 13 | 60.63 | 5.05 | 28.49 | 1.57 |
| 19 | 14 | 68.67 | 5.28 | 25.31 | 1.75 |
| 20 | 16 | 64.06 | 4.27 | 24.71 | 1.63 |
| 21 | 13 | 61.22 | 5.1 | 29.82 | 2.35 |
| 22 | 22 | 68.5 | 3.26 | 24.06 | 1.1 |
| 23 | 10 | 62.08 | 6.9 | 21.47 | 2.38 |
| 24 | 13 | 47.29 | 3.94 | 20.12 | 1.43 |
| Total | 371 | 1496.28 | 4.51 | 676.74 | 1.75 |

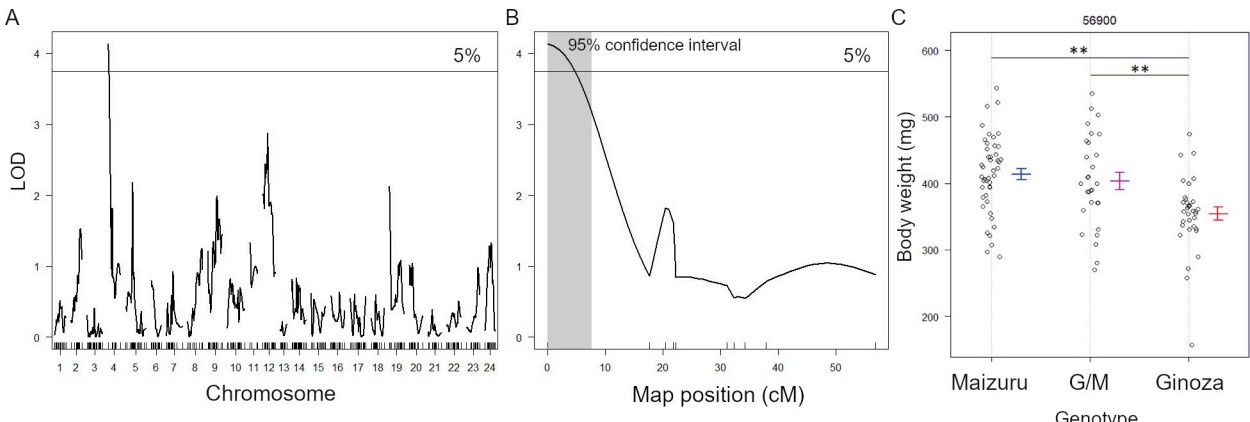

**Fig 4. QTL analysis of the F₂ generation.** (A) Results for all chromosomes. The significant QTL on chromosome 4 (B) has a peak LOD of 4.14. Gray areas indicate 95% Bayesian CI. (C) Body weight for genotypes of SNP marker 56900, which had the highest LOD score among the markers. Points represent the body weight of the F₂ individuals homozygous for the Maizuru alleles (n = 29), heterozygous for the Ginoza and Maizuru alleles (G/M, n = 46), and homozygous for the Ginoza alleles (n = 34). The means and standard errors (SE) are presented for each genotype. Differences between the mean body weight of the Ginoza and G/M genotypes and of the Ginoza and Maizuru fish were statistically significant (*p < 0.01, ANOVA and Tukey's honestly significance difference (HSD) test) but those between the G/M and Maizuru fish were not.

associated with body weight in broilers [43]; *scly* (selenocysteine lyase), *cep19* (19-kDa centrosomal protein), *dhcr24* (24-dehydrocholesterol reductase), and *lmo4* (LIM domain transcription factor) are associated with body weight and obesity traits in mammals [44–47]; and *mrpl55* (mitochondrial ribosomal protein L55) has a critical role in development and body size in *Drosophila* [48]. Furthermore, *sgta* (small glutamine-rich tetratricopeptide repeat-containing protein alpha) is a regulator of growth hormone receptors, which consequently influence body weight, because *sgta* knockout mice are smaller than wild-type mice [49].

Among the identified candidate genes, nine had amino acid substitutions that distinguished Maizuru from Ginoza (Table 2). The SNED1 amino acid substitution G1013S was predicted to be deleterious (functionally different) according to the PROVEAN score. In addition, the Ginoza amino acid variant, glycine, was prevalent among other vertebrate groups, whereas the serine residue in Maizuru was not found in other investigated species (Fig 5). SNED1 in humans is known as insulin-responsive sequence DNA-binding protein 1 (IRE-BP1); it is a transcription factor involved in the determination of BMI [37] and the activation of insulin-responsive genes and obesity [50]. *SNED1* is located at the terminal region of chromosome 2 in humans. Patients with a deletion in that region, with breakpoints at or within cytogenetic band 2q37, have a short stature among other features [51]. Sned1 might be involved in mouse skeletal development, as *Sned1* knockout mice have craniofacial malformations and growth defects [52]. Therefore, we speculate that a functional difference in SNED1 protein activity caused by the amino substitution that distinguished Maizuru from Ginoza induced the difference in body weight. However, unidentified genes in the QTL region might also influence body weight through mechanisms other than differences in protein function, such as changes in their expression levels and/or profiles. These speculations await further investigation.

Future studies are necessary to identify the genetic variation(s) responsible for the differences in body weight between the medaka populations studied here. Nevertheless, the present results will contribute to the marker-assisted selection of economically important aquaculture species and provide a better understanding of the genetic mechanisms underlying ecological differences in body weight among populations at different latitudes.

**Table 2. SNPs with non-synonymous substitutions in the candidate genes in Maizuru and Ginoza.**

| Gene symbol (Accession no.)[1] | Description | Position of mRNA (bp)[1] | SNP | | | Number of amino acids | Non-synonymous substitutions | | |
|---|---|---|---|---|---|---|---|---|---|
| | | | Position (bp)[1] | Maizuru | Ginoza | | Position (aa)[1] | Maizuru | Ginoza |
| *sgcb* (XP_020558516.1) | sarcoglycan beta | 1356188–1364186 | 1360124 | G | C | 296 | 98 | Leu | Val |
| *cilp2* (XP_011471792.1) | cartilage intermediate layer protein 2 | 2139202–2161322 | 2161297 | T | C | 1302 | 3 | Lys | Arg |
| | | | 2161159 | G | A | | 49 | Ser | Leu |
| | | | 2149453 | T | A | | 146 | Thr | Ser |
| | | | 2149400 | A | C | | 163 | Asp | Glu |
| | | | 2149359 | C | T | | 177 | Ser | Asn |
| | | | 2147355 | T | C | | 326 | Asp | Gly |
| | | | 2147167 | C | T | | 389 | Val | Ile |
| | | | 2146872 | C | T | | 420 | Gly | Asp |
| | | | 2141205 | T | A | | 961 | Tyr | Phe |
| | | | 2141068 | T | A | | 1007 | Met | Leu |
| | | | 2140229 | G | C | | 1286 | Ile | Met |
| *mef2b* (XP_011471858.1) | myocyte-specific enhancer factor 2B | 2306973–2324496 | 2314502 | G | A | 421 | 218 | Pro | Leu |
| | | | 2314431 | A | C | | 242 | Ser | Ala |
| | | | 2313743 | C | T | | 266 | Gly | Ser |
| | | | 2308851 | A | C | | 298 | Val | Gly |
| | | | 2308220 | G | C | | 360 | Ser | Thr |
| | | | 2308128 | T | C | | 391 | Ile | Val |
| *rfxank* (XP_011471878.1) | DNA-binding protein RFXANK | 2358970–2362447 | – | – | – | 208 | – | – | – |
| *rab6b* (XP_004067647.1) | ras-related protein Rab-6B | 2693698–2739982 | – | – | – | 215 | – | – | – |
| *sned1* (XP_011471983.1) | sushi, nidogen and EGF-like domain-containing protein 1 | 2824005–2850954 | 2850480 | T | C | 1349 | 39 | Lys | Glu |
| | | | 2843533 | T | G | | 207 | Gln | Pro |
| | | | 2842008 | T | A | | 372 | Thr | Ser |
| | | | 2841981 | A | G | | 381 | Tyr | His |
| | | | 2837261 | A | G | | 559 | Leu | Pro |
| | | | 2837260 | C | T | | | | |
| | | | 2837250 | C | T | | 563 | Ala | Thr |
| | | | 2835940 | T | G | | 660 | Asn | Thr |
| | | | 2835939 | G | T | | | | |
| | | | 2835923 | C | G | | 666 | Val | Leu |
| | | | 2835883 | G | C | | 679 | Thr | Ser |
| | | | 2835439 | G | C | | 772 | Gln | Glu |
| | | | 2835283 | A | G | | 792 | Tyr | His |
| | | | 2833621 | T | G | | 912 | Lys | Gln |
| | | | 2833038 | T | C | | 1013 | Ser | Gly |
| | | | 2830096 | T | G | | 1070 | Thr | Pro |
| | | | 2829960 | A | G | | 1115 | Val | Ala |
| | | | 2829735 | C | A | | 1146 | Ala | Ser |
| | | | 2829731 | C | T | | 1147 | Arg | Lys |
| | | | 2829460 | C | G | | 1213 | Ser | Thr |
| | | | 2825995 | T | A | | 1315 | Gln | Leu |
| | | | 2825713 | T | A | | 1346 | Asn | Ile |

*(Continued)*

**Table 2.** (Continued)

| Gene symbol (Accession no.)[1] | Description | Position of mRNA (bp)[1] | SNP | | | Number of amino acids | Non-synonymous substitutions | | |
|---|---|---|---|---|---|---|---|---|---|
| | | | Position (bp)[1] | Maizuru | Ginoza | | Position (aa)[1] | Maizuru | Ginoza |
| *scly* (XP_023809718.1) | selenocysteine lyase | 2908002–2933175 | 2908316 | A | C | 445 | 411 | Ser | Ala |
| *cep19* (XP_023809746.1) | centrosomal protein 19 kDa | 3244247–3245808 | 3244607 | C | T | 157 | 128 | Asp | Asn |
| *dhcr24* (XP_004067663.1) | delta(24)-sterol reductase | 3287533–3295716 | 3288635 | A | C | 516 | 396 | Ser | Ala |
| | | | 3288401 | G | A | | 452 | Ala | Val |
| *lmo4* (XP_023809796.1) | LIM domain transcription factor LMO4 | 4586958–4596199 | – | – | – | 165 | – | – | – |
| *sgta* (XP_023809802.1) | small glutamine-rich tetratricopeptide repeat-containing protein alpha | 5201953–5215418 | – | – | – | 342 | – | – | – |
| *mrpl55* (XP_023809802.1) | 39S ribosomal protein L55, mitochondrial | 5446241–5450258 | 5446451 | T | C | 138 | 16 | Thr | Met |
| | | | 5446578 | C | A | | 21 | Pro | Thr |
| | | | 5446708 | T | A | | 64 | Leu | Gln |
| | | | 5450099 | T | A | | 118 | Ser | Thr |

[1]Genes and positions are according to the NCBI ASM223467v1 (GCF_002234675.1) assembly.

The G1013S substitution in SNED1 with a deleterious effect (significantly different function) estimated by PROVEAN is shaded in gray.

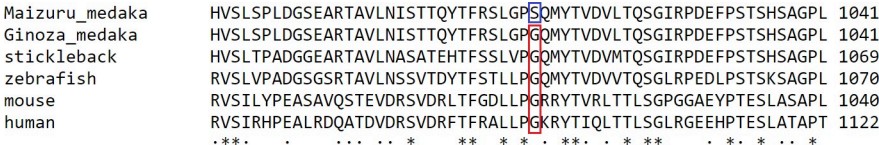

**Fig 5. SNED1 protein sequence comparison.** Glycine at position 1013 in Ginoza SNED1 (red square) is conserved among other vertebrate species. Serine (blue square) was found only in Maizuru SNED1.

## Supporting information

**S1 Table. Single nucleotide polymorphism (SNP) markers used for quantitative trait locus (QTL) mapping and genotyping for each SNP in the parental, F$_1$, and F$_2$ generations.** Number of markers used for QTL mapping: 371. (XLSX)

**S2 Table. Single nucleotide polymorphisms (SNPs) on the coding regions in 12 candidate genes from the Maizuru and Ginoza individuals.** Number of Maizuru and Ginoza fish: n = 4 each. (XLSX)

**S3 Table. Genotyping amplified fragment length polymorphism marker ch4_0.68M and SNP marker 56900 in the parental and F$_2$ generations.** The forward and reverse primer sequences of ch4_0.68M for genotyping using PCR are 5′-caattgcctgtttgtcagttacac-3′ and 5′-cgcctaatgccactccagcac-3′, respectively. Their locations are 685055–685078 bp and 685136–685156 bp on chromosome 4, respectively. The sizes of the amplified fragments separated using Microchip Electrophoresis System MCE®-202 MultiNA microchip electrophoresis (Shimadzu Corporation, Kyoto, Japan). (XLSX)

## Acknowledgments

We thank the National Bio-Resource Project Medaka of MEXT, Japan, and the Data Integration and Analysis Facility of NIBB, Japan, for permitting the use of their facilities. We thank Dr. T. Shimmura, D. Adachi, N. Baba, A. Akama, and M. Okubo for technical assistance.

## Author Contributions

**Funding acquisition:** Takashi Yoshimura, Ai Shinomiya.

**Investigation:** Tamiris I. Yassumoto, Mana Nakatsukasa, Atsushi J. Nagano, Masaki Yasugi, Takashi Yoshimura, Ai Shinomiya.

**Writing – original draft:** Tamiris I. Yassumoto, Ai Shinomiya.

**Writing – review & editing:** Takashi Yoshimura.

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
