## [Decision Letter · Decision Letter 0]

24 Mar 2020

PONE-D-20-01808

Genetic analysis of body weight in wild populations of medaka fish from different latitudes

PLOS ONE

Dear Dr. Yoshimura,

Thank you for submitting your manuscript to PLOS ONE. After careful consideration, we feel that it has merit but does not fully meet PLOS ONE’s publication criteria as it currently stands. Therefore, we invite you to submit a revised version of the manuscript that addresses the points raised during the review process.

We would appreciate receiving your revised manuscript by May 08 2020 11:59PM. To enhance the reproducibility of your results, we recommend that if applicable you deposit your laboratory protocols in protocols.io, where a protocol can be assigned its own identifier (DOI) such that it can be cited independently in the future. For instructions see: http://journals.plos.org/plosone/s/submission-guidelines#loc-laboratory-protocols

We look forward to receiving your revised manuscript.

Kind regards,

Christos Maravelias, Ph.D.

Academic Editor

PLOS ONE

Journal Requirements:

2. We note that you are reporting an analysis of a microarray, next-generation sequencing, or deep sequencing data set. PLOS requires that authors comply with field-specific standards for preparation, recording, and deposition of data in repositories appropriate to their field. Please upload these data to a stable, public repository (such as ArrayExpress, Gene Expression Omnibus (GEO), DNA Data Bank of Japan (DDBJ), NCBI GenBank, NCBI Sequence Read Archive, or EMBL Nucleotide Sequence Database (ENA)). In your revised cover letter, please provide the relevant accession numbers that may be used to access these data. For a full list of recommended repositories, see http://journals.plos.org/plosone/s/data-availability#loc-omics or http://journals.plos.org/plosone/s/data-availability#loc-sequencing.

"This work was supported in part by a JSPS KAKENHI “Grant-in-Aid for Scientific Research (C)” (15K07163, 19K06785) to AS and “Grant-in-Aid for Specially Promoted Research” (26000013) and “Grant-in-Aid for Scientific Research (S)” (19H05643) and by the Human Frontier Science Program (RGP0030/2015) to TY. The funders had no role in study design, data collection and analysis, decision to publish, or preparation of the manuscript."

Reviewers' comments:

Reviewer's Responses to Questions

**Comments to the Author**

1. Is the manuscript technically sound, and do the data support the conclusions?

Reviewer #1: Yes

Reviewer #2: Partly

2. Has the statistical analysis been performed appropriately and rigorously? 

Reviewer #1: Yes

Reviewer #2: No

3. Have the authors made all data underlying the findings in their manuscript fully available?

Reviewer #1: Yes

Reviewer #2: No

4. Is the manuscript presented in an intelligible fashion and written in standard English?

Reviewer #1: Yes

Reviewer #2: No

5. Review Comments to the Author

Reviewer #1: The articile is well-written, and the topic quite interesting with potential implications for aquaculture breeding programmes.

No major modifications are required prior to publication.

The only minor modification suggested is related to the introduction.

In fact, while the authors recognized that their results could have implications for aquaculture, it seems not strictly related to their research the reference to human diseases and medicine science in the two paragraphs (lines 53-64). This part could be shortned or even omitted.

Reviewer #2: I am unfortunately confused with the way the authors approached this subject.

The content of the paper does not help much the readers to understand how the laboratory experiments and the downstream analyses were performed.

More specifically:

- in the abstract, the authors have at least a paragraph for obesity research. How is this relevant to the manuscript? Using medaka as a model for human health? We already know that most of these traits are polygenic. Therefore, the results of your study pointing on a single gene for growth might need more thorough reanalysis.

- For the RAD approach: did you finally use both or a single enzyme? The number of SNPs you finally used (371) is much fewer than that people usually report for RAD or ddRAD approaches. You do not comment on this.

- Mapping and QTL identification: with the above mentioned number of SNPs it is normal to have a very sparse linkage map with an average of only 15 markers per linkage group or chromosome. To my opinion, this number is very small to have a sound analysis.

- On chromosome 4, there are only 11 markers mapped and the one showing a positive association with growth is located at the extreme; from the literature we know that sometimes it might be erroneous.

- The 12 genes span a region, if I am not mistaken, of approximately 4Mb and the authors have invested a lot of effort and expenses to go for sequencing all these genes for the “source” populations (how many fish exactly?) and identify fixed differences. Very little is reported for the way you performed it; the primers, how many PCR reactions and different fragments per gene etc.

6. PLOS authors have the option to publish the peer review history of their article (what does this mean?). If published, this will include your full peer review and any attached files.

Reviewer #1: No

Reviewer #2: No

---

## [Author Response · Author response to Decision Letter 0]

9 May 2020

Response to Reviewer 1

The article is well-written, and the topic quite interesting with potential implications for aquaculture breeding programs. No major modifications are required prior to publication. The only minor modification suggested is related to the introduction. In fact, while the authors recognized that their results could have implications for aquaculture, it seems not strictly related to their research the reference to human diseases and medicine science in the two paragraphs (lines 53-64). This part could be shortened or even omitted. 

Response: We have shortened the two paragraphs as suggested (lines 50–57).

Responses to Reviewer 2

Is the manuscript presented in an intelligible fashion and written in standard English?

No

Response: We used the service of an English proofreading company to improve clarity. 

I am unfortunately confused with the way the authors approached this subject.

The content of the paper does not help much the readers to understand how the laboratory experiments and the downstream analyses were performed.

Response: We have added more details to facilitate a better understanding of the experimental procedures and downstream analyses.

1.- in the abstract, the authors have at least a paragraph for obesity research. How is this relevant to the manuscript? Using medaka as a model for human health? We already know that most of these traits are polygenic. Therefore, the results of your study pointing on a single gene for growth might need more thorough reanalysis. 

Response: We have deleted the word “medical” from the abstract and have diminished the claim in the Introduction section. We have shortened and revised the two paragraphs that described obesity research (lines 50–57). Furthermore, we have added a description of the relevance and relationships among body weight, growth, and animal health, which affect aquaculture economics. 

2.- For the RAD approach: did you finally use both or a single enzyme? The number of SNPs you finally used (371) is much fewer than that people usually report for RAD or ddRAD approaches. You do not comment on this.

Response: We used both enzymes; we have added the relevant information in the Materials and Methods section (line 124). 

We initially obtained 5,019 RAD-sequencing markers that were genotyped in > 70% of all 155 individuals from the parental, F1, and F2 generations. 

In this experiment, F2 individuals originated from five parental Maizuru and Ginoza pairs. To obtain informative and reliable markers for QTL analysis, we selected markers with genotyping datasets in the parental, F1, and F2 generations with homozygous but different genotypes between all Maizuru and all Ginoza parents and a heterozygous genotype in F1. We also selected markers that were genotyped in > 80% of the F2 fish. After these selection processes, we reduced the number of markers to 371 for QTL analysis.

3. - Mapping and QTL identification: with the above mentioned number of SNPs it is normal to have a very sparse linkage map with an average of only 15 markers per linkage group or chromosome. To my opinion, this number is very small to have a sound analysis.

Response: Indeed, more markers would increase the statistical power to detect more QTL and be useful for high-resolution mapping. However, we decided to use only reliable markers to minimize or avoid errors. 

4. - On chromosome 4, there are only 11 markers mapped and the one showing a positive association with growth is located at the extreme; from the literature we know that sometimes it might be erroneous.

Response: We designed and genotyped the new marker “ch4_0.68M,” which is located on the distal region of the medaka chromosome 4. The genetic distance between ch4_0.68M and the RAD 56900 marker supported the linkage of these two loci. Therefore, the position of 56900 and the QTL on chromosome 4 is reliable. Genotyping data on the new marker can be found in the supplemental S3 Table and in the revised manuscript (lines 209–211). 

5. - The 12 genes span a region, if I am not mistaken, of approximately 4Mb and the authors have invested a lot of effort and expenses to go for sequencing all these genes for the “source” populations (how many fish exactly?) and identify fixed differences. Very little is reported for the way you performed it; the primers, how many PCR reactions and different fragments per gene etc.

Response: We sequenced the whole genomes of four Maizuru and four Ginoza individuals using HiSeq 4000 (Illumina). Since these whole genome data will be published elsewhere (in preparation), the data cannot be submitted to a public database at this time. However, we showed all the SNP in the coding regions of the 12 genes in all Maizuru and Ginoza individuals included in the variant analyses in the supplemental S2 Table, lines 158–160. After reanalyzing the SNP of these Maizuru and Ginoza individuals, we added more data to Table 2.

---

## [Decision Letter · Decision Letter 1]

3 Jun 2020

Genetic analysis of body weight in wild populations of medaka fish from different latitudes

PONE-D-20-01808R1

Dear Dr. Yoshimura,

We’re pleased to inform you that your manuscript has been judged scientifically suitable for publication and will be formally accepted for publication once it meets all outstanding technical requirements.

Kind regards,

Christos Maravelias, Ph.D.

Academic Editor

PLOS ONE

Additional Editor Comments (optional):

Reviewers' comments:

Reviewer's Responses to Questions

**Comments to the Author**

1. If the authors have adequately addressed your comments raised in a previous round of review and you feel that this manuscript is now acceptable for publication, you may indicate that here to bypass the “Comments to the Author” section, enter your conflict of interest statement in the “Confidential to Editor” section, and submit your "Accept" recommendation.

Reviewer #2: All comments have been addressed

2. Is the manuscript technically sound, and do the data support the conclusions?

Reviewer #2: Yes

3. Has the statistical analysis been performed appropriately and rigorously? 

Reviewer #2: Yes

4. Have the authors made all data underlying the findings in their manuscript fully available?

Reviewer #2: Yes

5. Is the manuscript presented in an intelligible fashion and written in standard English?

Reviewer #2: Yes

6. Review Comments to the Author

Reviewer #2: the authors have adequately addressed the comments raised in the previous round of review and I feel that this manuscript is now acceptable for publication

7. PLOS authors have the option to publish the peer review history of their article (what does this mean?). If published, this will include your full peer review and any attached files.

Reviewer #2: No

---

## [Editor Report · Acceptance letter]

5 Jun 2020

PONE-D-20-01808R1 

Genetic analysis of body weight in wild populations of medaka fish from different latitudes 

Dear Dr. Yoshimura:

I'm pleased to inform you that your manuscript has been deemed suitable for publication in PLOS ONE. Congratulations! Your manuscript is now with our production department. 

Kind regards, 

on behalf of

Dr. Christos Maravelias 

Academic Editor

PLOS ONE